# Temporal Patterns of Air Leak Resolution in Secondary Spontaneous Pneumothorax: A Hazard Function Analysis for Optimal Intervention Timing

**DOI:** 10.3390/jcm14114003

**Published:** 2025-06-05

**Authors:** Ryo Takeyama, Yoshikane Yamauchi, Shinya Kohmaru, Shizuka Morita, Hikaru Takahashi, Tomoki Nishida, Yuichi Saito, Yukinori Sakao

**Affiliations:** Department of Surgery, School of Medicine, Teikyo University, Tokyo 173-8605, Japanyuichi.saito@med.teikyo-u.ac.jp (Y.S.);

**Keywords:** secondary spontaneous pneumothorax, air leak, hazard function, chest tube drainage

## Abstract

**Objectives**: This study was aimed to identify risk factors for persistent air leak after chest tube placement for secondary spontaneous pneumothorax and to determine the optimal timing of treatment. **Methods**: We retrospectively analyzed 221 cases of secondary spontaneous pneumothorax in patients aged ≥50 years who were treated with chest tube drainage. Patients were categorized into the observation group or additionally treated group based on whether they received interventional treatment beyond chest tube drainage. Air leak resolution patterns were analyzed using hazard function analysis. Risk factors were evaluated using univariate and multivariate analyses. **Results**: Hazard function analysis revealed that the probability of air leak resolution decreased by approximately 50% within the first 5 days after the initiation of chest tube drainage, with only 33% of cases resolving by day 7. Beyond days 7–10, resolution probability stabilized at a minimal level. Multivariate analysis identified previous pneumothorax history (HR: 0.422, *p* = 0.007) and low geriatric nutritional risk index (GNRI) (HR: 2.521, *p* < 0.001) as significant independent risk factors for persistent air leak. Further analysis of early resolution (within 7 days) identified female sex (HR: 0.24, *p* = 0.003), absence of previous pneumothorax (HR: 0.21, *p* = 0.003), and higher GNRI values (HR: 1.04, *p* = 0.008) as positive predictors. **Conclusions**: Risk stratification based on pneumothorax history and nutritional status enables the optimization of the timing of intervention for persistent air leak. We recommend considering additional treatment between days 7 and 10 of chest tube drainage, with earlier intervention for high-risk patients. This approach may improve patient outcomes while avoiding unnecessarily prolonged conservative management.

## 1. Introduction

Secondary spontaneous pneumothorax (SSP) is a pneumothorax caused by an underlying lung disease [1]. While chronic obstructive pulmonary disease (COPD) accounts for 50% to 70% of cases, other causes include pulmonary tuberculosis (40%) and malignancy (8%) [2,3]. SSP typically develops when the underlying lung pathology induces the rupture of subpleural cystic air spaces (blebs or bullae) [4].

Compared with primary spontaneous pneumothorax, SSP is associated with higher rates of complications, recurrence, and mortality [3,5], mainly because SSP patients already have compromised lung function and are typically in worse general condition. Mortality is significantly higher in SSP patients (4.6%) than in all patients with spontaneous pneumothorax (1.7%), with emphysema being the most frequent cause of death [6]. In Japan, 90% of elderly pneumothorax patients have underlying lung disease [7], which frequently makes the condition refractory to treatment, leading to prolonged hospitalization and decreased activities of daily living.

SSP treatment recommendations vary among the different treatment guidelines. Although the 2010 British Thoracic Society (BTS) guidelines recommend treatments predominantly based on pneumothorax size [8], the more recent 2023 BTS [9] and 2024 Joint ERS/EACTS/ESTS guidelines [10] are less definitive about alternatives to chest tube drainage. As a result, chest tube drainage remains the predominant treatment. However, the management of patients with SSP after chest tube placement is challenging, as over 20% of patients require prolonged chest drainage exceeding 15 days [6,11]. While the BTS guidelines recommend surgical consultation after 48 h [8], many patients are poor candidates for surgery because of compromised pulmonary function, and specific criteria for intervention timing remain unclear, although surgical intervention has shown favorable outcomes in selected elderly patients [12,13].

Management approaches for SSP vary among institutions and have evolved over time. At our institution, treatment decisions have generally followed contemporary thoracic society guidelines, though specific protocols varied during our study period as institutional practices evolved. The heterogeneity in treatment approaches and the lack of standardized criteria for intervention timing motivated this retrospective analysis to identify objective factors that could guide more evidence-based decision-making. Therefore, this study exploratively investigates the clinical factors related to the duration of chest tube drainage for patients with SSP to determine the need for additional treatment and the optimal timing of additional therapeutic intervention, if required. Hopefully, our findings will help guide clinicians in treatment decision making and improve patient outcomes.

## 2. Materials and Methods

### 2.1. Ethics Statement

This study received institutional review board approval (no. 23-065, approved on 29 August 2023) and was performed in accordance with the principles of the Declaration of Helsinki. The requirement for informed consent was waived owing to the retrospective nature of the study.

### 2.2. Subjects

This study retrospectively reviewed pneumothorax cases in patients aged ≥50 years who underwent chest tube drainage at our hospital from April 2010 to March 2023. The age limit of 50 years was selected because secondary spontaneous pneumothorax predominantly affects older patients with underlying lung disease, and this threshold helps identify a population with distinct risk characteristics compared to younger patients with primary pneumothorax. The study period (April 2010 to March 2023) represents the era during which comprehensive electronic medical records were maintained at our institution, ensuring adequate data availability for analysis.

Chest tube insertion was performed by board-certified thoracic surgeons, board-certified pulmonologists, or senior residents (≥3rd year) under direct attending supervision using standardized institutional protocols. Tube placement was confirmed by immediate post-insertion chest radiography in all cases. Given the retrospective nature of this study spanning 13 years, treatment approaches and clinical decision-making evolved during this period. While general institutional practices followed established thoracic society guidelines, specific criteria for chest tube insertion, timing of additional interventions, and surgical referrals were not strictly protocolized and varied among attending physicians and over time. This clinical heterogeneity reflects real-world practice patterns and motivated our analysis to identify objective factors that could guide more standardized decision-making.

At our institution, decisions regarding additional interventional treatment for persistent air leak were made during daily morning conferences, considering multiple factors, including air leak persistence, the patient’s overall condition, comorbidities, social circumstances, and treatment preferences. While additional treatment was commonly considered around days 7–10 of chest tube drainage, the actual timing varied considerably based on individual patient factors and clinical circumstances. Some patients received earlier intervention due to medical urgency or social factors, while others were managed conservatively for longer periods based on patient preference, comorbidities, or gradual improvement in air leak volume. This variability in clinical decision-making, reflecting real-world practice patterns, motivated our analysis to identify more objective criteria that could guide intervention timing.

Cases with air leak < 24 h, iatrogenic pneumothorax, or those caused by trauma/malignancy were excluded, as were cases with unknown air leak cessation dates. For patients with multiple pneumothorax episodes during the study period, only episodes occurring ≥3 months after the complete resolution of the previous episode were included, ensuring clinical independence between events. The collection of nutritional and inflammatory parameters was based on growing literature demonstrating their prognostic value in respiratory diseases, though the specific parameters collected varied somewhat over the study period as clinical practice evolved. Collected data included demographics, medical history (smoking, steroid use), comorbidities (interstitial pneumonia, asthma, chronic obstructive pulmonary disease), clinical parameters (Body Mass Index (BMI), C-reactive protein-to-albumin ratio (CAR), geriatric nutritional risk index (GNRI), prognostic nutritional index (PNI), CRP–albumin–lymphocyte index (CALLY index)), bullae distribution, Goddard Score, and treatment details.

The Goddard score is a visual evaluation method of radiologic emphysematous changes on CT that was previously reported in scientific literature [14,15]. It evaluates emphysematous changes by dividing the lung field into six areas (upper, middle, and lower on each side) and scoring each area from 0–4 based on emphysema severity, with a maximum possible score of 24. CT images were obtained using specific parameters (120 kVp, 200 mA, 1–2 mm section thickness). For each case, Goddard scores were calculated using CT images taken within 3 months before pneumothorax occurrence or after sufficient lung expansion following chest tube placement. While some cases had missing Goddard scores, measurements were obtained for over 80% of cases.

### 2.3. Statistical Analysis

Statistical analyses were performed using Prism 8 (GraphPad Software, Inc., San Diego, CA, USA) and EZR (Saitama Medical Center, Jichi Medical University, Saitama, Japan) [16], a graphical user interface for R version 4.4.2 (The R Foundation for Statistical Computing, Vienna, Austria). Continuous variables were presented as median and interquartile range, and *t*-tests were performed using Welch’s method. The optimal cutoff values for continuous prognostic indexes were determined using the web-based application Cutoff Finder (https://molpathoheidelberg.shinyapps.io/CutoffFinder_v1/, accessed on 5 January 2025) [17], which fits Cox proportional hazard models to the dichotomized variable. The optimal cutoff was defined as the point with the most significant split.

Cumulative distribution function curves were constructed and compared using the log-rank test. To identify factors associated with air leak resolution, univariate analyses were first performed using the log-rank test. Variables with *p* < 0.05 were entered into a multivariate analysis. For multivariate analysis, a Cox proportional hazards model was used when considering time to resolution, while logistic regression was used when time-course information was not available. *p* < 0.05 was considered significant.

The hazard function, defined as the instantaneous risk of the event of interest occurring within a fairly narrow timeframe [18], was used to model the probability of air leak cessation as a function of chest drainage time. The time scale was discretized in 1-day increments, and all hazard rates for cessation were measured as “events/patients at risk per 1-day interval”. The hazard function was calculated using the following formula:(1)$$h(t)=\frac{Number\;of\;air\;leak\;cessation\;cases\;in\;a\;day}{Number\;of\;cases\;with\;air\;leak\;just\;before\;the\;time}$$

Scatterplots of cessation rate over time were constructed, and the locally weighted scatter plot smoothing method used to create a smooth curve [19]. One-day instantaneous hazard rates were estimated because the rate estimates were unstable due to random fluctuation, and a smooth curve is more useful for understanding the hazard rate patterns.

Since nutrition–inflammation indicators (such as BMI, CAR, GNRI, PNI, and CALLY index) may correlate with each other, correlation analysis was performed to assess multicollinearity. As strong correlations were observed among these indicators, only the factor showing the strongest association in univariate analysis was included in the multivariate analysis as a representative indicator of nutritional and inflammatory status. The Goddard score was included to evaluate the impact of structural changes in the lung parenchyma on persistent air leakage.

## 3. Results

### 3.1. Patient Characteristics and Clinical Outcomes According to Treatment Groups

Among 417 cases of pneumothorax treated using chest tube drainage during the study period, 221 cases in 195 patients were included for analysis after applying criteria. Among the 221 cases analyzed, 195 individual patients were included, with 26 cases (11.8%) representing recurrent pneumothorax episodes. All recurrent episodes occurred at least 3 months after complete resolution of the previous episode, ensuring that each episode represented an independent clinical event rather than treatment failure or incomplete resolution. A study flowchart is shown in Figure 1. The air leak was managed with chest tube drainage alone in 97 cases (O-group); 124 cases required additional treatment (A-group).

Patient characteristics stratified by treatment approach are presented in Table 1. The O-group had a slightly older median age compared to the A-group (73.0 vs. 70.5 years, *p* = 0.067), though this difference did not reach statistical significance. Males predominated in both groups, with a higher proportion in the A-group (83.1% vs. 77.3%, *p* = 0.308). Both groups demonstrated comparable smoking history, with median pack-years of 40 and 44 in the O-group and A-group, respectively (*p* = 0.364), and similar rates of current smokers (15.2% vs. 20.0%, *p* = 0.470). Notably, the A-group had a significantly higher prevalence of previous pneumothorax episodes (40.3% vs. 15.5%, *p* < 0.001), with ipsilateral recurrence being the most common pattern (29.0% vs. 13.4%). The distribution of comorbidities, including interstitial pneumonia (15.5% vs. 15.3%, *p* = 1.000), asthma (9.3% vs. 9.7%, *p* = 1.000), and COPD (41.2% vs. 39.0%, *p* = 0.782), was comparable between groups. Regarding anatomical factors, bilateral bullae distribution was more frequent in the A-group (74.2% vs. 63.9%, *p* = 0.156), though not reaching statistical significance. The median Goddard score, reflecting emphysema severity, was similar between groups (7 vs. 8, *p* = 0.318). Inflammation–nutrition indicators showed significant differences between the groups. The CAR was significantly higher in the O-group (0.20 vs. 0.11, *p* = 0.002), while the CALLY index was significantly lower (0.54 vs. 1.28, *p* = 0.003). Other nutritional parameters, including BMI, GNRI, and PNI, showed no significant differences between groups.

### 3.2. Temporal Patterns of Air Leak Resolution

To comprehensively understand the natural course of air leak resolution in pneumothorax patients, we conducted a time-to-event analysis on the entire cohort. For this analysis, cases that received additional interventional treatment were appropriately censored at the exact time point when such interventions were implemented, thereby preserving the integrity of the natural resolution data. Figure 2a presents the cumulative incidence curve of air leak cessation with 95% confidence intervals (shown as dotted lines). The curve demonstrates a biphasic pattern characterized by a steep initial increase in resolution rate during the first week post-drainage, with approximately 33% of all cases achieving complete air leak cessation within 7 days. This early resolution phase is followed by a more gradual increase in cumulative resolution over the subsequent weeks, with the curve approaching but not exceeding 75% by day 30. The instantaneous rate of air leak cessation is further elucidated in Figure 2b, which depicts the hazard function over time. This analysis reveals a pronounced non-linear pattern in resolution dynamics. Initially, the air leak cessation rate is at its maximum (approximately 0.08 per day) immediately following chest tube insertion, indicating that the probability of spontaneous resolution is highest during this early period. However, this rate rapidly decreases by approximately 50% within the first 5 days of drainage, followed by a continued but more gradual decline until approximately day 7–10, where the function reaches an inflection point at around 0.025 per day. After this critical juncture, the hazard function stabilizes and shows minimal further reduction, suggesting a constant but substantially lower probability of spontaneous resolution for persistent air leaks beyond this timeframe.

### 3.3. Predictive Factors for Air Leak Cessation Without Additional Intervention

Table 2 presents the comprehensive statistical analysis of factors associated with air leak cessation without requiring additional interventional procedures. We conducted both univariate (log-rank test) and multivariate (Cox proportional hazards regression) analyses to identify independent predictors of spontaneous resolution. In the univariate analysis, seven factors demonstrated statistically significant associations with air leak cessation: advanced age (>84 years old, *p* < 0.001), previous history of pneumothorax (*p* = 0.009), Goddard score (*p* = 0.044), BMI (*p* = 0.022), CAR (*p* = 0.029), GNRI (*p* = 0.005), and PNI (*p* = 0.014). Several other factors approached but did not reach statistical significance, including male sex (*p* = 0.081) and smoking history exceeding 17 pack-years (*p* = 0.089). Notably, current smoking status, immuno-suppressive therapy, and comorbidities, including interstitial pneumonia, asthma, and COPD, did not demonstrate significant associations with air leak cessation in this cohort.

Prior to conducting the multivariate analysis, we performed a correlation analysis among the inflammation–nutrition indicators (BMI, CRP/albumin ratio, GNRI, and PNI) to address potential multicollinearity issues. These parameters showed substantial intercorrelation, as expected given their overlapping biochemical and physiological foundations. Based on both statistical strength in the univariate analysis (lowest *p*-value) and established clinical relevance in respiratory conditions, GNRI was selected as the representative inflammation–nutrition indicator for inclusion in the multivariate model. This approach allowed us to avoid statistical instability while preserving the essential nutritional component in our final model.

The multivariate Cox proportional hazards regression analysis identified two independent factors significantly associated with air leak cessation. Previous history of pneumothorax emerged as a negative predictor (HR 0.422, 95% CI 0.225–0.791, *p* = 0.007), indicating that patients with recurrent pneumothorax had a 57.8% lower likelihood of spontaneous resolution compared to those experiencing their first episode. Conversely, GNRI > 105.2 was identified as a strong positive predictor (HR 2.521, 95% CI 1.459–4.357, *p* < 0.001), with patients above this threshold having a 2.5-fold higher probability of spontaneous air leak cessation. Advanced age and Goddard score, while significant in univariate analysis, did not maintain independent statistical significance in the multivariate model (*p* = 0.115 and *p* = 0.112, respectively), suggesting their effects may be partially mediated through nutritional status or other included variables.

The analyses of factors associated with cessation of air leak are shown in Table 2. Univariate analysis revealed that significantly associated factors were age, previous history of pneumothorax, Goddard score, BMI, CAR, GNRI, and PNI. However, because BMI, CAR, GNRI, and PNI are all inflammation–nutrition indicators and are highly correlated, only GNRI was included in the multivariate analysis as a representative of inflammation–nutrition indicators, as it showed a stronger association in the univariate analysis. The multivariate analysis revealed several factors independently associated with prolonged air leak. Patients with a previous pneumothorax showed significantly lower likelihood of cessation (Hazard ratio: 0.422, *p* = 0.007). Additionally, the analysis showed that GNRI score was significantly associated with air leak cessation, with higher GNRI scores indicating higher rates of spontaneous air leak cessation (*p* < 0.001).

### 3.4. Temporal Dynamics of Spontaneous Air Leak Resolution in the Observation Group

Figure 3a illustrates the cumulative probability of air leak cessation over time in the O-group, with the solid line representing the point estimate and the dotted lines demarcating the 95% confidence intervals. The curve demonstrates a distinctly steeper gradient compared to the overall cohort analysis (Figure 2a), with three notable phases of resolution: an initial rapid phase during the first 2–3 days post-drainage where approximately 35% of air leaks resolved, followed by a moderately steep phase between days 3 and 7, where an additional 15% resolved (cumulative ~50% by day 5), and a third, more gradual phase extending from days 7 to 15, during which another 20–25% of cases achieved resolution. By day 15, the cumulative probability of spontaneous resolution plateaued at approximately 75–80%, suggesting that air leaks persisting beyond this timeframe have a diminished likelihood of resolving without intervention.

The corresponding hazard function analysis in Figure 3b reveals critical insights into the instantaneous rate of air leak cessation in the O-group. The initial hazard rate was substantially higher than that observed in the overall cohort (0.19 vs. 0.08 per day), indicating that patients who ultimately achieved spontaneous resolution without intervention had a notably higher early probability of air leak cessation. This rate demonstrated a continual decline over the first week, decreasing to approximately 50% of its initial value by day 7 (0.10 per day). Subsequently, the hazard function stabilized at approximately 0.08 per day between days 8 and 12, forming a plateau that differed from the pattern observed in the overall cohort.

### 3.5. Predictors of Early Spontaneous Air Leak Cessation Within the Critical 7-Day Window

Table 3 presents the results of our comparative analysis between patients who experienced spontaneous air leak cessation within 7 days (*n* = 62) and those with persistent air leaks beyond this timeframe (*n* = 123). For this analysis, we excluded patients who received additional interventions within the first 6 days after chest tube insertion to prevent treatment-related confounding and to specifically examine factors associated with the natural resolution process.

In the univariate analysis, three factors demonstrated statistically significant associations with early air leak cessation: female sex (30.6% of resolved cases vs. 17.1% of persistent cases, *p* = 0.039), absence of previous pneumothorax (85.5% of resolved cases vs. 67.5% of persistent cases, *p* = 0.009), and higher GNRI (median 94.62 vs. 90.52, *p* = 0.042). Interestingly, traditional clinical factors such as age, smoking history, current smoking status, comorbidities (interstitial pneumonia, asthma, COPD), and emphysema severity (Goddard score) showed no significant association with early air leak cessation in this cohort. Subsequent multivariate logistic regression analysis confirmed and further quantified three independent predictors of early air leak cessation. Female sex demonstrated a stronger protective effect compared with male sex (*p* = 0.003). Similarly, previous history of pneumothorax emerged as a significant negative predictor (*p* = 0.003). GNRI remained significant in the multivariate model (*p* = 0.008).

## 4. Discussion

In this study of 221 pneumothorax cases, we performed a novel time-course analysis of air leak cessation using the hazard function, which has not been performed previously. Hazard function analysis provides deeper insights than cumulative incidence or survival analyses by quantifying the instantaneous probability of air leak resolution at any given time point. This approach revealed distinct patterns of resolution dynamics, with a critical transition period during the first 10 days. Beyond providing cumulative resolution rates, our analysis uniquely demonstrated the dynamic changes in resolution probability over time, offering clinicians a more nuanced understanding of when air leak resolution becomes less likely. Furthermore, our multivariate analysis identified previous pneumothorax history and GNRI as factors strongly associated with prolonged air leak, while female sex, no previous pneumothorax history, and higher GNRI were significantly associated with air leak resolution by day 7. The GNRI was initially used to assess the nutritional status of elderly patients with chronic disease [20]. To the best of our knowledge, our study is the first to demonstrate its association with air leak cessation in SSP.

In a 1995 time-course study, air leak resolved within 10 days of chest tube placement in approximately 40% of cases of spontaneous pneumothorax in a diseased lung; in cases in a normal lung, >90% resolved within the same period [21]. In our study of SSP, air leak resolution occurred by day 10 in 33% of cases, which is a similar result; however, our cumulative frequency curve is inversely oriented compared to their survival curve.

Our hazard function analysis showed that the probability of air leak resolution decreases by half after day 5 and diminishes substantially around days 7 through 10. The cumulative frequency curve for the spontaneous resolution group alone also suggests that cases with a persistent air leak beyond 10 days are unlikely to achieve spontaneous resolution and will likely require additional treatment—approximately 80% of the air leaks in this group resolved within the first 10 days of chest tube placement. This suggests that expecting spontaneous resolution beyond day 10 may be unrealistic and that additional treatment should probably be strongly considered between days 7 and 10. Furthermore, for patients with a history of pneumothorax and/or with relatively low GNRI score, additional treatment might be considered, even at the time of initial chest tube placement, because spontaneous resolution by day 7 is not highly anticipated in these patients. On the other hand, the fact that the Goddard score did not remain as a significant factor suggests that spontaneous resolution in SSP patients is not greatly influenced by lung parenchymal findings. This implies that treatment plans should be considered without being overly concerned about imaging findings, even when strong emphysematous changes are observed radiographically.

While quantitative air leak volume measurements were not systematically recorded in our study, the existing literature demonstrates the importance of standardized air leak assessment in secondary spontaneous pneumothorax. Higher volumes of air leaking through the chest tube generally correlate with prolonged air leak duration. For example, in postoperative settings, an air leak greater than 50 mL/min predicted persistent air leak lasting more than 72 h, indicating that higher initial leak volumes delay healing [22]. Furthermore, Kida et al. in 2021 [23] demonstrated that interventions reducing pneumothorax volume and air leak scores significantly improved outcomes in inoperable SSP patients. Pneumothorax volumes decreased from 29.1% ± 17.3% to 12.1% ± 8.8%, and air leak scores (Cerfolio classification) dropped from 4.0 ± 2.0 to 1.0 ± 1.5 after bronchial occlusion and pleurodesis, correlating with an 85% success rate in chest tube removal [23]. The integration of standardized air leak volume assessment with our identified risk factors (pneumothorax history and GNRI) could potentially enhance the accuracy of prognostic models for secondary spontaneous pneumothorax.

This study has several limitations. Recall bias may have been present because of the study’s retrospective design. In addition, the decision to proceed with additional treatment for an air leak was not standardized, so operator bias may have been present as well. Furthermore, quantitative air leak volume data were not available and therefore not analyzed. While qualitative assessments of air leak volume were available, these are of questionable reliability and are susceptible to observer bias. Additionally, point-of-care ultrasound assessment was not routinely employed during our study period, though emerging evidence suggests that ultrasound evaluation of pleural sliding and lung expansion could provide additional objective data for air leak assessment and prognostic evaluation. Given that clinical decisions are frequently based on air leak volume in real-world clinical practice, the risk factors for persistent air leak identified in this study should be interpreted with caution. However, it is important to emphasize that our findings maintain significant clinical relevance despite these limitations. All data analyzed in this study were derived from actual chest tube placement records, providing concrete clinical evidence. The identified risk factors enable clinicians to anticipate cases with a higher likelihood of persistent air leak without necessarily waiting for air leak volume measurements. This proactive risk assessment allows for earlier strategic treatment planning, which represents a valuable contribution to clinical practice. Therefore, we believe this study offers meaningful insights that can enhance clinical decision making in the management of pneumothorax. Future prospective studies with standardized intervention protocols are needed to validate our findings. The integration of standardized air leak volume assessment and ultrasound evaluation with our identified risk factors could potentially enhance the accuracy of prognostic models in future prospective studies.

The clinical significance of this research lies in providing objective data to support risk stratification in SSP patients. Our hazard function analysis demonstrates that the probability of spontaneous air leak resolution becomes minimal beyond days 7–10, providing evidence-based support for the commonly practiced timing of intervention consideration around this timeframe. However, our findings suggest that risk stratification based on pneumothorax history and GNRI could enable more individualized timing decisions. While actual intervention timing will continue to vary based on individual patient circumstances, social factors, and patient preferences, our risk stratification approach could help inform these decisions by providing prognostic information to both patients and clinicians. This evidence-based framework could complement clinical judgment and enhance patient counseling regarding expected clinical course.

While prospective validation is required, our findings suggest potential approaches to risk-stratified management:


(a)Patients with favorable prognostic factors (first-time pneumothorax, GNRI > 105.2)
-May benefit from expectant management for 7–10 days given the higher observed resolution rates in our cohort.-Additional intervention could be considered if air leak persists beyond this timeframe.-Patient counseling can include information about relatively favorable prognosis for spontaneous resolution.(b)Patients with unfavorable prognostic factors (recurrent pneumothorax, GNRI ≤ 105.2)-Should be counseled about the lower probability of spontaneous resolution observed in our analysis.-May warrant closer monitoring with consideration of earlier intervention timing around days 7–10.-Could benefit from early discussion of alternative treatment options and realistic expectation setting.


Given the limitations discussed above, these findings provide a foundation for future prospective studies to validate risk-stratified management approaches in SSP patients.

## 5. Conclusions

Our hazard function analysis of air leak resolution patterns in SSP patients revealed critical temporal dynamics: the probability of spontaneous resolution decreases significantly after day 5 and becomes minimal beyond days 7–10. We identified previous pneumothorax history and low GNRI score as significant risk factors for persistent air leak. These findings support implementing early risk stratification based on these factors, enabling clinicians to make more informed decisions about additional treatment timing. This approach could potentially improve patient outcomes by avoiding unnecessarily prolonged conservative management in cases with low probability of spontaneous resolution. Our study provides a more nuanced understanding of air leak resolution dynamics that can guide clinical decision-making for pneumothorax management.

## Figures and Tables

**Figure 1 jcm-14-04003-f001:**
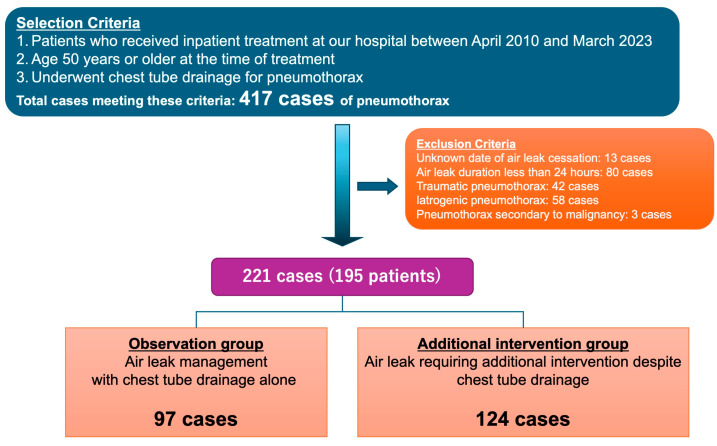
Study flowchart.

**Figure 2 jcm-14-04003-f002:**
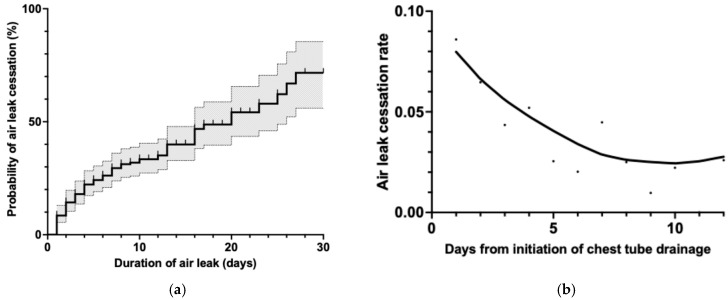
Air leak resolution patterns in all secondary spontaneous pneumothorax cases. (**a**) Cumulative incidence curve of air leak cessation showing approximately 33% resolution by day 7 after chest tube placement. Cases that received additional treatment were censored at the time of intervention. The shaded area represents the 95% confidence interval. (**b**) Smoothed hazard function analysis demonstrating the rate of air leak cessation over time. Note the approximately 50% decrease in cessation rate within the first 5 days after chest tube placement, followed by a gradual decline until reaching a plateau around days 7–10.

**Figure 3 jcm-14-04003-f003:**
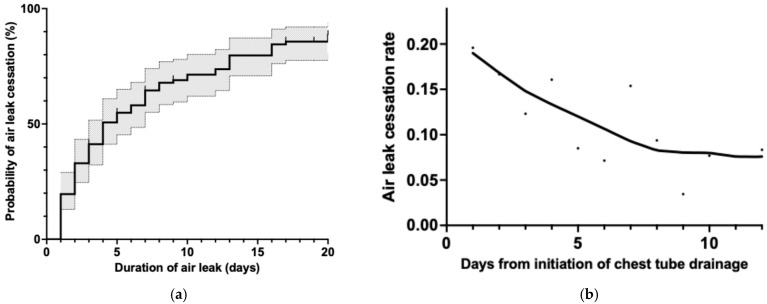
Air leak resolution patterns in cases with spontaneous resolution (O-group). (**a**) Cumulative incidence curve of air leak cessation showing approximately 50% resolution within the first 5 days and 70% resolution by day 10 after chest tube placement. The shaded area represents the 95% confidence interval. (**b**) Smoothed hazard function analysis demonstrating the rate of air leak cessation over time. Note the decline in cessation rate by approximately 50% around day 7, followed by stabilization at a lower rate between days 8 and 10.

**Table 1 jcm-14-04003-t001:** Patients’ characteristics.

Factors		Observation Group (O-Group, *n* = 97)	Additionally Treated Group (A-Group, *n* = 124)	*p*-Value
Age (years)		73.0 [64, 80]	70.5 [63, 77]	0.067
Sex (F/M)		22:75	21:103	0.308
Smoking history (pack-year)		40 [11, 60]	44 [20, 70]	0.364
Current smoker (%)		14 (15.2)	24 (20.0)	0.470
Comorbidity of interstitial pneumonia (%)		15 (15.5)	19 (15.3)	1.000
Comorbidity of asthma (%)		9 (9.3)	12 (9.7)	1.000
Comorbidity of COPD (%)		40 (41.2)	48 (39.0)	0.782
Previous history of pneumothorax (%)	None	82 (84.5)	74 (59.7)	<0.001
	Ipsilateral	13 (13.4)	36 (29.0)	
	Contralateral	0 (0.0)	6 (4.8)	
	Bilateral	2 (2.1)	8 (6.5)	
Distribution of bullae (%)	None	20 (20.6)	13 (10.5)	0.156
	Local	10 (10.3)	15 (12.1)	
	Ipsilateral	4 (4.1)	4 (3.2)	
	Bilateral	62 (63.9)	92 (74.2)	
Steroid (>10 mg prednisolone) administration (%)		4 (4.1)	10 (8.1)	0.276
Immunosuppressant administration (%)		2 (2.1)	3 (2.4)	1.000
Goddard score		7 [4, 10]	8 [6, 12]	0.318
BMI (kg/m^2^)		21.08 [17.67, 23.33]	19.88 [17.48, 22.46]	0.172
CAR		0.20 [0.09, 1.41]	0.11 [0.03, 0.41]	0.002
GNRI		90.94 [79.38, 106.25]	94.88 [85.46, 101.74]	0.553
PNI		41.96 [32.52, 49.98]	43.52 [37.84, 49.70]	0.106
CALLY index		0.54 [0.05, 2.15]	1.28 [0.21, 4.94]	0.003
Spontaneously cured without additional interventions (%)		86 (88.7)	0 (0.0)	<0.001
Observation period without additional interventions (days)		4.00 [2.00, 10.00]	10.00 [7.00, 14.00]	<0.001

Continuous variables are presented as median [interquartile range] due to the non-normal distribution of the data.

**Table 2 jcm-14-04003-t002:** Uni- and multivariate analysis of factors associated with spontaneous air leak resolution.

Factors	Log-Rank Test	Cox Hazard Regression	
	*p*-Value	Hazard Ratio (95% CI)	*p*-Value
Age (>84 years old)	<0.001	1.988 (0.846–4.675)	0.115
Sex (male)	0.081		
Smoking history (>17 pack-year)	0.089		
Current smoker	0.279		
Previous history of pneumothorax	0.009	0.422 (0.225–0.791)	0.007
Steroid (>10 mg prednisolone) administration	0.229		
Immunosuppressant administration	0.949		
Comorbidity of interstitial pneumonia	0.923		
Comorbidity of asthma	0.869		
Comorbidity of COPD	0.562		
Distribution of bullae	0.17		
Goddard score	0.044	1.703 (0.883–3.285)	0.1123
BMI (>20.8 kg/m^2^)	0.022		
CRP/albumin (>0.025)	0.029		
GNRI (>105.2)	0.005	2.521 (1.459–4.357)	<0.001
PNI (>53.5)	0.014		
CALLY index	0.105		

**Table 3 jcm-14-04003-t003:** Uni- and multivariate analysis of factors associated with spontaneous air leak resolution within 7 days after chest tube insertion.

Factors		Univariate Analysis			Logistic Regression	
		Resolved (*n* = 62)	Unresolved (*n* = 123)	*p*-Value	Hazard Ratio (95% CI)	*p*-Value
Age (years old)		73.0 [50, 96]	73.0 [50, 96]	0.321		
Sex (F/M)		19:43	21:102	0.039	0.24 (0.09–0.62)	0.003
Smoking history (pack-year)		40.00 [0, 129]	41.00 [0, 186]	0.389		
Current smoker (%)		10 (16.7)	22 (19.0)	0.837		
Comorbidity of interstitial pneumonia (%)		9 (14.5)	21 (17.1)	0.833		
Comorbidity of asthma (%)		7 (11.3)	11 (8.9)	0.608		
Comorbidity of COPD (%)		26 (41.9)	46 (37.4)	0.632		
Previous history of pneumothorax (%)		9 (14.5)	40 (32.5)	0.009	0.21 (0.08–0.58)	0.003
Distribution of bullae (%)	None	6 (9.7)	14 (11.4)	0.186		
	Local	3 (4.8)	4 (3.3)			
	Ipsilateral	37 (59.7)	88 (71.5)			
	Bilateral	15 (24.2)	17 (13.8)			
Steroid (>10 mg prednisolone) administration (%)		2 (3.2)	12 (9.8)	0.146		
Immunosuppressant administration (%)		0 (0.0)	5 (4.1)	0.170		
Goddard score		8.00 [0.00, 18.00]	7.00 [0.00, 19.00]	0.519		
BMI (kg/m^2^)		21.37 [12.78, 34.29]	19.53 [11.77, 31.83]	0.055		
CRP/albumin		0.14 [0.00, 8.53]	0.19 [0.00, 20.10]	0.904		
GNRI		94.62 [51.37, 126.56]	90.52 [48.51, 119.90]	0.042	1.04 (1.01–1.06)	0.008
PNI		44.37 [14.16, 83.34]	41.88 [11.80, 64.12]	0.320		
CALLY index		0.74 [0.00, 82.08]	0.54 [0.00, 155.76]	0.901		

Continuous variables are presented as median [interquartile range] due to the non-normal distribution of the data.

## Data Availability

Data are available upon request due to privacy and ethical restrictions.

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
