# Peer review of "Temporal Patterns of Air Leak Resolution in Secondary Spontaneous Pneumothorax: A Hazard Function Analysis for Optimal Intervention Timing"

_jcm, 2025, doi:10.3390/jcm14114003_

Round 1
Reviewer 1 Report
Comments and Suggestions for Authors
Thank you to the authors for this work.
Thank you to the editor for this wonderful review opportunity.
Here are my comments:
In the introduction, could you clarify your criteria for drain placement in the case of a primary pneumothorax.
Also in the introduction, specify what you consider patients ineligible for surgery due to impaired pulmonary function; please clarify your criteria.
Please specify whether the placement of the drains was codified. Are you sure that all drains were correctly placed? By whom? Experience? Did you have a way to verify the placement protocol and the correct position of the drain? Please clarify and rephrase. Were all drains placed according to the same placement criteria (inclusion, exclusion, indication)?
Among your patient selection criteria, why are you limited to 50 years of age? Why did you limit the period from April 2010 to March 2023?
Can you justify your exclusion criteria? Can you justify the collection of nutritional data based on the literature?
In your results, you specify that the 221 cases come from 195 patients; therefore, there are patients who had more than one intervention. What should be done with these patients? Can a repeat intervention change the results? Please clarify or rephrase.
There seems to be a missing comma on line 226. Is that correct?
Could you elaborate more clearly on the protocol you could propose based on the type of patient, the amount of leakage (is there an influence of the initial amount of leakage; please also specify) on the resolution time? And your different items? Are they based on the number of days after chest drain insertion?
Finally, are there cases of pneumothorax without criteria for chest drain insertion that nevertheless require them over time? If so, based on what criteria and over what time frame? Is it possible to specify or formulate a rule regarding the management of simple pneumothoraces?
Were you able to determine what type of secondary treatment was necessary, specifically? Did it depend on certain criteria? Did you have your own criteria for implementing secondary treatment? What were the results of these secondary treatments in terms of effectiveness and time to complete resolution after initiating secondary treatment?
Thank you very much.
Reviewer 2 Report
Comments and Suggestions for Authors
Good afternoon,
First of all, thank you for the opportunity to review this article. I found it very interesting.
Secondly, I would like to say that the manuscript is very well prepared. As a humble suggestion, I would recommend including a section on the study’s limitations, as well as outlining potential directions for future research.
Thirdly, did you consider including ultrasound/POCUS as factors to be taken into account in the study?
Thank you very much.
Kind regards,
Round 2
Reviewer 1 Report
Comments and Suggestions for Authors
Thank you to the authors for this proofreading.
Thank you to the editor for their trust.
The comments were widely followed, and all questions were answered.
The focus was on the real goal of the study, which is precisely to attempt to determine the timing of a second course of therapy in spontaneous secondary pneumothorax.
Bravo
And thank you.